# Invited Review: Increasing Milk Yield and Negative Energy Balance: A Gordian Knot for Dairy Cows?

**DOI:** 10.3390/ani13193097

**Published:** 2023-10-04

**Authors:** Holger Martens

**Affiliations:** Institute of Veterinary Physiology, Free University of Berlin, Oertzenweg 19b, 14163 Berlin, Germany; holger.martens@fu-berlin.de

**Keywords:** β-hydroxybutyric acid, dairy cow, fatty liver, ketosis, lipidosis, negative energy balance, non-esterified fatty acids, production diseases

## Abstract

**Simple Summary:**

Dairy cows have been primarily selected during the last century for higher milk production with no attention being paid to other traits such as a sufficient dry matter intake for the augmented milk requirement. The delay between the rapid increase in milk yield and dry matter intake causes a gap called the negative energy balance. A growing body of evidence suggests that this gap increases with any enhancement in milk production. The missing energy (and protein) is covered by the mobilization of non-esterified fatty acids from subcutaneous and abdominal fat stores and to a small extent by the release of amino acids from muscle. Unfortunately, the mobilization of non-esterified fatty acids is greater than the metabolic capacity of the cow, leading to an increase in this metabolite in the blood. The surplus being ectopically deposited in muscle and in the liver, the uptake overwhelms the metabolic capability of the liver, resulting in the production and release of β-hydroxybutyric acid and the resynthesis of non-esterified fatty acids to triglycerides. The limited export of triglycerides causes an accumulation of these compounds, with the consequence of fatty liver or lipidosis, which clearly causes subclinical and clinical ketosis. Furthermore, lipidosis is associated with various so-called “production diseases”, including inflammation, oxidative, endoplasmatic stress and immunosuppression. Hence, the coupling of more milk with insufficient dry matter intake is the key to understanding lipidosis, ketosis and other health risks in dairy cows postpartum.

**Abstract:**

The continued increase in milk production during the last century has not been accompanied by an adequate dry matter intake (DMI) by cows, which therefore experience a negative energy balance (NEB). NEB is low and of minor importance at low milk yield (MY), such as for the nutrition of one calf, and under these circumstances is considered “natural”. MY and low DMI around parturition are correlated and are the reason for the genetic correlation between increasing MY and increasing NEB up to 2000 MJ or more for 2–3 months postpartum in high-genetic-merit dairy cows. The extension and duration of NEB in high-producing cows cannot be judged as “natural” and are compensated by the mobilization of nutrients, particularly of fat. The released non-esterified fatty acids (NEFAs) overwhelm the metabolic capacity of the cow and lead to the ectopic deposition of NEFAs as triglycerides (TGs) in the liver. The subsequent lipidosis and the concomitant hampered liver functions cause subclinical and clinical ketosis, both of which are associated with “production diseases”, including oxidative and endoplasmatic stress, inflammation and immunosuppression. These metabolic alterations are regulated by homeorhesis, with the priority of the physiological function of milk production. The prioritization of one function, namely, milk yield, possibly results in restrictions in other physiological (health) functions under conditions of limited resources (NEB). The hormonal framework for this metabolic environment is the high concentration of growth hormone (GH), the low concentration of insulin in connection with GH-dependent insulin resistance and the low concentration of IGF-1, the so-called GH-IGF-1 axis. The fine tuning of the GH-IGF-1 axis is uncoupled because the expression of the growth hormone receptor (GHR-1A) in the liver is reduced with increasing MY. The uncoupled GH-IGF-1 axis is a serious impairment for the GH-dependent stimulation of gluconeogenesis in the liver with continued increased lipolysis in fat tissue. It facilitates the pathogenesis of lipidosis with ketosis and, secondarily, “production diseases”. Unfortunately, MY is still increasing at inadequate DMI with increasing NEB and elevated NEFA and beta–hydroxybutyric acid concentrations under conditions of low glucose, thereby adding health risks. The high incidences of diseases and of early culling and mortality in dairy cows are well documented and cause severe economic problems with a waste of resources and a challenge to the environment. Moreover, the growing public concerns about such production conditions in agriculture can no longer be ignored.

## 1. Introduction

The milk production of dairy cows has been continuously increased during the last century and will probably be raised in the future. The increase is the result of the genetic selection for more milk, proper animal management and the provision of an adequate environment. Yields of 10.000 kg milk per lactation (305 d) or even more are not unusual under intensive production conditions. This development has been carefully outlined by Baumgard et al. [1] for the USA, in which milk yield (MY) has increased from ca. 2.500 kg in 1950 to 10.000 kg in 2016. Similarly, in Germany, a rise has been reported from 2.600 kg in 1950 to 8557 kg in 2022 as the mean for all breeds [2]. This rapid expansion in production “will be recognized as the “Golden Age” of lactation biology” [3].

Nevertheless, dairy cows experience a negative energy balance (NEB) postpartum (p.p.) which is compensated by the mobilization of some energy stores and their later replenishment during the lactation period. This cyclic change is normal [4], although excessive and long-lasting mobilization p.p. can lead to health problems, such as ketosis, the impairment of fertility or so-called “production diseases” [5]. A clear-cut transition from physiology to pathophysiology is difficult to discern.

The aim of the current overview is to present the hypothesis that increasing milk production is associated with an inadequate dry matter intake (DMI) leading to a rising NEB. The discrepancy between input (DMI) and output (MY) is covered by the mobilization, primarily, of energy (fat). Hence, postpartum MY, DMI, mobilization and NEB represent traits that guarantee the nutrition of the calf, on the one hand, even under tough environmental conditions, whereas increasing MY under intensive farming conditions is associated, on the other hand, with an inadequate DMI and, consequently, with an expansion of the extent and duration of NEB. The cascade behind these events will be described here and includes (a) DMI during transition; (b) milk yield and NEB; (c) mobilization, metabolites and hormones; (d) the pathogenesis of lipidosis as a precondition of ketosis and its association with “production diseases”; and, finally, (e) the possible genetic background.

It includes (a) insufficient dry matter intake during transition, (b) milk yield and negative energy balance, (c) the mobilization of reserves with alterations of hormones and metabolites, and, finally, (d) the association between NEFAs, lipidosis and diseases. This is not a systematic review. Rather, it is an approach aimed at improving the understanding of the physiology (biology) of the dairy cow during the transition period at low milk yield and of possible metabolic and hormonal “derailments” during increasing milk production at insufficient DMI and the accompanying NEB, together with its undoubted health risks.

## 2. Dry Matter Intake during Transition

During the last 15–10 d antepartum (a.p.), DMI continuously decreases and is 20–40% lower at the day of parturition. The decline is more pronounced in overconditioned cows [6], and DMI a.p. is genetically correlated with DMI p.p. [7]. The drop at parturition is slowly compensated by increasing DMI p.p. until a maximum intake is achieved between 8 and 12 weeks. MY grows much faster, with a peak at 6–8 weeks, resulting in a gap between the requirement for milk production and DMI with a subsequent NEB. These cyclic changes with a lag between milk production and DMI p.p. have been studied in the past with no satisfactory explanation [6] and are still poorly understood [8]. Knight [9] has summarized these uncertainties as follows: “Perhaps, the real conundrum is why so much effort is exerted by agricultural nutritionists and dairy farmers in trying to persuade the early-lactation cow to eat more. She knows better!” An explanation for this assumption “She knows better” has been suggested by Vernon and Pond [10]: “The inappetance around parturition is probably a throwback to the wild state when mothers would need to remain at the nest for a period of time and would be unable to feed”. Hence, the increase in milk production at insufficient DMI with the subsequent NEB has a probable biological and possible genetic background for ensuring the nutrition and survival of the calf. The amount of milk for the nutrition of one calf is limited, as is the NEB, and is a negligible challenge for the metabolism and health of the cow.

This biological situation, namely, low MY and minor NEB, has been altered by the selection for higher milk production and frequent milking for maximal MY at a remaining inadequate DMI. The changed shape of the lactation curve has noticeably contributed to this situation, as Gravert [11] has stated: “The curve of feed intake (p.p. the author) still corresponds to the non-domesticated cow in according with the requirement of the suckling calf while the lactation curve has been altered by artificial selection for higher milk yields”.

The possible drawbacks of the insufficient DMI have been known for decades. Balch [12] emphasized this problem in the introduction to a meeting about feed intake in dairy cows: “Feed intake regulation: A limiting factor in animal production (the dairy cow, the author)”. This shortcoming was expressed again in the 1980s: “…that selection of milk yield would not automatically increase feed intake of dairy cows in the first part of lactation” [13]. Subsequently, Arendonk et al. [14] pointed out: “Intake capacity might be considered as an additional trait in the selection goal to avoid an increase of negative energy balance in early lactation.” Some 23 years (!) later, von Leesen et al. [15] proposed an improvement in the breeding of dairy cows: “Thus, an energy balance (indicator) should be included in future breeding programs”, which again has been underlined by Rodehutscord and Titze [16]: “A remaining goal for reducing the energy deficit is a higher ranking of DMI in the breeding index.” However, the new German breeding index for Holstein cows does not include this trait, because the measurement of DMI is difficult and the corresponding data for genetic selection are not sufficiently available. This is, to the knowledge of the author, a worldwide problem.

## 3. Milk Yield and Negative Energy Balance

An increasing MY requires an augmented DMI because the requirement for maintenance is exceeded by a factor of 3–5 within a few weeks p.p. [17] and is accompanied by a heavy metabolic load, as expressed by Sheldon et al. [18]: “For a typical dairy cow producing 40 L of milk/d, the metabolic energy requirements for milk production are about 200 MJ/d, whereas only about 65 MJ/d is needed for maintenance. An equivalent metabolic demand for humans is running 3 marathons per day”. Hence, the dairy cow is a “marathon runner”, but over a time-span of weeks.

The rapid change in metabolism and its metabolic load require comprehensive regulation, which has been named homeorhesis, in order to provide “orchestrated changes for priorities of a physiological state” [19], but it has the important consequence that milk production (“physiological state”) takes priority [19] over other physiological functions that are then possibly restricted [19,20,21,22]. Obviously, feed intake does not belong to the priority of milk production. On the contrary, “feed intake during early lactation was negatively correlated with milk yield” [23], a finding later confirmed by Karacaören et al. [24] and by Liinamo et al. [25]. Similarly, Krattenmacher et al. [26] observed a moderate negative genetic correlation between MY and energy balance in early lactation, making understandable the early conclusion of Buttchereit et al. [27]: “that continued selection for high milk production will lead to a further increase in the postpartum energy deficit, unless energy balance is directly or indirectly included in the breeding programs with appropriate economic weights”.

The production-dependent increase in NEB was demonstrated decades ago. Brand et al. [28] measured an NEB of 1445 MJ_NEL_ (the mean of all cows) with a duration of 11 weeks p.p. in the first lactation of German Holstein Friesian cows and correlated the NEB with MY: MY < 25 kg/d = 580 MJ_NEL_ NEB, 25–30 kg/d = 1323 MJ_NEL_ NEB and >30 kg/d = 1956 MJ_NEL_ NEB. This MY-dependent increase in NEB was confirmed by Friggens et al. [29] by a comparison of MY and the nadir of NEB in Jersey, Danish Red and Danish Holstein cows. The nadir (MJ/d) was augmented at higher MY: Danish Holstein > Danish Red > Jersey. Insufficient DMI p.p. was further demonstrated in an international co-operation (Austria, Germany, Switzerland) by Gruber et al. [30]. The increase in DMI p.p. was 0.1 kg DM/kg milk, clearly indicating an increasing deficit with increasing MY.

As mentioned above, MY p.p. is partly uncoupled from DMI, a characteristic that has a biological and probable genetic background. The possible benefits are that the nutrition of the calf is independent of DMI and independent of the search for food by the cow, which can then concentrate her time on supervising her newborn calf. Even a shortage of food under tough environmental conditions does not challenge the nutrition of the calf. The deficit is compensated by the mobilization of reserves. Since the requirement (and the appetite) of the calf is limited, nutrition at the cost of the mother is rarely a challenge for the metabolism or health of the cow. Data from Hart et al. [31] support this assumption: British Hereford cows with milk production only for the nutrition of the calf exhibit a gain in bodyweight (BW) p.p., which nevertheless does not mean a lack of BW loss, because the increase in DMI and thus gut fill might compensate lost BW. However, the NEB in beef cattle (natural) is of minor importance in comparison with that of dairy cows, and the magnitude can only be estimated, with a range of perhaps 100–200 MJ. In contrast, Hart et al. (1974) detected a diminished BW in British Holstein dairy cows for ca. 10 weeks p.p. in their study [31]. Hence, the current NEB is a man-made problem in the modern high-genetic-merit dairy cow with its higher MY. The situation can be considered as “natural” in Hereford or beef cattle, but not in high-producing dairy cows.

The genetic trait of milk production is uncoupled from sufficient DMI and, hence, the continuous selection for more milk has exacerbated the NEB now to 2000 MJ or more with a duration of 2–3 months [32,33], a situation that “exhibits characteristics of chronic under-nutrition” [20]. The 2000 MJ lost represent ca. one ΔBCS (5-point), 80 kg BW or 40–50 kg fat (39.7 MJ/kg) and ca. 10 mm fat thickness on the cow’s back [34]. An amount of 1 kg mobilized BW includes 64% fat, 8% protein and 28% water [35].

Hence, the current NEB in high-producing cows is far above the original (biological) loss of BW but is still often considered as “natural”, because weight loss in mammals after parturition is well known; this biological event is thus frequently used to declare the NEB of the dairy cow as “normal”. For example, Bauman [36] and Horst et al. [37] mention the heavy weight loss in bears, seals and whales p.p. However, these animals belong to species with “adaptive fasting” [38] and do not eat at all for weeks or months p.p. They totally rely on their reserves during the time spent raising their offspring. This strategy is obviously optimal for the nutrition of their young and can hardly be compared with that of the dairy cow. Moreover, the milk of the above species does not contain glucose [39] and no incidence of ketosis has been reported in these animals. Large amounts of glucose are required for the synthesis of lactose in cow’s milk [40], and glucose is without doubt the metabolic “eye of the needle” during early lactation. In dairy cows, glucose is mainly produced by gluconeogenesis, which can scarcely be maintained without an adequate DMI.

## 4. Mobilization of Reserves: Hormones and Metabolites

The rapid and massive changes in the metabolism in the cow pp. are accompanied by a plethora of alterations in various hormones, e.g., leptin, prolactin, ghrelin, resistine, corticoid, thyroxine, adiponectin and FGF21. Furthermore, an increase in growth hormone (GH) and a decrease in insulin (Ins) attended by insulin resistance (InsR) and reduced IGF-1 concentrations occur, which together are highly likely to determine the observed metabolic adaptation. GH and insulin accompanied by InsR and IGF-1 heavily influence metabolism p.p.; major metabolic effects are summarized in Table 1.

High GH, low Ins and InsR, and low IGF-1 are the “orchestrated (hormonal) changes for priorities of a physiological state” [19], i.e., milk production p.p. during a period of NEB, and lead to enhanced lipolysis and the release of NEFAs for energy production (ATP) and milk fat synthesis, reduced lipogenesis, the stimulation of gluconeogenesis in the liver and the reduced uptake of glucose in muscle and fat tissue, with partitioning of the spared glucose for the synthesis of lactose in milk.

The combined effects of GH and the GH-dependent InsR are under the feedback control of IGF-1 within the so-called GH-IGF-1 axis. GH, via the growth receptor 1A in the liver, stimulates IGF-1 synthesis and its release into the blood [44]. IGF-1 reduces the extrusion of GH from the pituitary gland, and this negative feedback finally controls catabolic reactions. Unfortunately, the fine-tuning signal cascade of the GH-IGF-1 axis can be uncoupled by the reduced expression of GHR-1A in the liver of dairy cows [45], although not in Angus beef cattle [46]. The degree of uncoupling of the GH-IGF-1 axis evidently depends on MY, as demonstrated by Lucy et al. [47]. Noticeable differences of uncoupling were observed in a study of Holstein Friesen cows from USA and New Zealand (NZ): USA > NZ.

The interruption of the GH-IGF-1 axis signal cascade is caused by the reduced expression of GHR-1A in the liver, resulting in decreased IGF-1 and, because of the lack of feedback, increasing GH concentrations in the blood. The decrease in GHR-1A expression is probably related to the low insulin concentration p.p., because the infusion of insulin + glucose p.p. increases its expression and “recouples” the GH-IGF-1 axis [43]. Thus, high-producing and “uncoupled” dairy cows p.p. are characterized by high concentrations of GH and low concentrations of insulin and IGF-1 in the blood.

These alterations in hormones and metabolites and the possible health risks arising from them have been demonstrated in two publications [48,49], the data of which can be used as examples for the complex and voluminous adaptation and maladaptation that occur p.p.

Gross and Bruckmaier [48] retrospectively ranked the cows in their study according to the highest NEFA concentration in weeks 1 to 4 p.p. The animals with the 33% highest and 33% lowest NEFA concentrations were selected and compared (Table 2). Other than insulin, all the parameters p.p. were (at various degrees and times) significantly different in cows with the highest NEFA concentrations.

Gross and Bruckmaier [48] explained these alterations based on “a higher degree of uncoupling of somatotropic axis with higher concentration of GH at concomitantly lower concentrations of plasma IGF-1…”

The increase in β-hydroxybutyric acid (BHB) to >1.2 mmol/L in this study needs to be mentioned here, as it clearly indicates subclinical ketosis and is related to severe lipidosis of the liver [49]. The data in Table 2 support the homeorhetic priority of milk production at insufficient DMI, an intake that is even significantly lower in high-producing cows despite a higher requirement.

## 5. NEFAs, Lipidosis and Diseases

The rapid increase in NEFAs p.p. is the result of enhanced lipolysis overwhelming the metabolic capacity of the cow [50] for the production of energy (ATP) and milk fat. The surplus of NEFAs causes the ectopic deposition of triglycerides in the muscle and liver [51]. Arshad and Santos [49] have demonstrated an almost linear correlation between NEFA concentration in the blood and lipidosis of the liver. The details of the pathogenesis of lipidosis are given in the reviews of Gross [52] and Martens [53], as are the possible impairments in liver functions.

One important direct effect of lipidosis is the positive and linear correlation with BHB blood values [49] that explains the risk of subclinical or clinical ketosis. These associations suggest a sequence of NEB, NEFAs, lipidosis and, finally, ketosis. Hence, lipidosis precedes ketosis or vice versa: ketosis p.p. is always connected with lipidosis. Furthermore, the degree of lipidosis is associated with various clinical diseases (Table 3).

A morbidity of 36.2%, even in the group with a physiological TGs (triglycerides) of 2.5%, is high and can hardly be accepted as normal. Remarkably, all listed diseases other than displaced abomasum exhibit an increased incidence with the degree of lipidosis. A causal correlation probably exists between TGs and subclinical ketosis, because TGs and BHB are correlated [49]. The association between TGs and the other diseases does not indicate a clear-cut causality and supports the conclusion of Sundrum [54]: “Metabolic disorders in the transition period indicate that the dairy cows’ ability to adapt is overstressed”, which is probably also the reason for the numerical decrease in survival [49] and agrees with the correlation of hyperketonemia and an increased culling rate [55].

The association between subclinical hypocalcemia (SCH) and TGs (Table 3) possibly has a metabolic (energetic) background. Cows with SCH (≤2.14 mmol/L) had higher NEFA and BHB concentrations [56], and cows with hypocalcemia (<1.0 mmol/L ionized Ca) and high NEFAs on the day of calving had significantly more lipids in their liver at day 7 [57]. “Hypocalcemic cows appeared to experience a more severe energy balance on the day of calving” [57], suggesting “that SCH increases lipomobilization” [56], which predisposes the cows to lipidosis. Indeed, cows with hypocalcemia (<2.00 mmol/L) had low insulin and high glucose concentrations, which “support the observation that physiological Ca concentration are required for glucose stimulation of insulin secretion” [58]. Experimental SCH in non-pregnant and non-lactating cows reduces DMI [59], possibly worsening the NEB p.p.

Surprisingly, Arshad and Santos [49] did not observe differences in the parameter of reproduction, which is well known to be associated with lipidosis [60] and increased BHB concentrations [61]. Timed artificial insemination was used and “abolished many of the potential effects of hepatic lipidosis on rates of insemination, that influences the rate of pregnancy in dairy cows” [49].

The working hypothesis of high milk yield at insufficient DMI and NEB is supported by the production data of Arshad and Santos [49] (Table 4).

The slight and significant increase in MY during rising lipidosis is not accompanied by a corresponding DMI. The priority of MY occurs despite numerically lower DMI (Table 4) and increased health risks (Table 3); this homeorhetic priority underlines possible restrictions of other (health) functions. Importantly, in both studies [48,49], the cows with the highest MY had a significant [48] (Table 2) or numerically [49] lower DMI (Table 4) and a higher NEB (Table 2 and Table 4).

## 6. Limitations

The transition period p.p. is accompanied by a high incidence of “production diseases”, as evidenced by the myriad publications about these problems, which cannot be addressed in detail. The interested reader will find further information about the metabolic adaptation and changes of hormones in [36,62,63], about interactions between NEB and reproduction in [64], about inflammation in [65,66] and immunosuppression in [67,68], about oxidative and endoplasmic stress in [69], about damage to mitochondrial functions of the liver in [70], about “production diseases” in [5], and about lipidosis in [52,53].

## 7. Discussion

*NEB and milk yield:* Mammals and, of course, dairy cows exhibit a decline in DMI around parturition and experience a physiological or “normal” NEB p.p. The reasons for this drop in DMI are not well understood, but the inappetance around parturition guarantees the nutrition of the calf at inadequate DMI (10) and the survival of the species is improved. The possible NEB only for the nutrition of the calf is of minor importance, as the following approximation of intake and requirement shows.

The German Society of Nutrition [71] gives, for the end of pregnancy, the recommendation of a DMI of 38 MJ_NEL_/d for the maintenance of a cow with a BW of 650 kg and 18 MJ_NEL_/d for the growth of the calf, the uterus and the mammary gland: 56 MJ_NEL_/d. 5–10 kg/d milk require 16.5 or 33 MJ_NEL_ [71] and, in total, 54.5–71.0 MJ_NEL_/d for a cow with this MY for the nutrition of the calf. Even at a low DMI of 8 kg/d with an energy density of 6.0 MJ_NEL_/kg, the possible NEB would be negligible at −6.5 MJ/d at 5 kg/d MY or slightly higher at −23 MJ_NEL_/d at 10/d kg MY. The possible deficit will be reversed at the same DMI by an energy density of 7.0 MJ_NEL_ to + 1.5 MJ_NEL_/d at 5 kg/d MY or will be reduced to −17 MJ_NEL_/d at 10 kg/d MY. These virtual calculations show that the NEB is low and probably transient for a few days p.p. at low MY. It is turned into a positive energy balance with increasing DMI p.p., because the MY will slowly increase with the appetite of the calf. Hence, MY according to the requirement of the calf hardly challenges the metabolism or health of a cow. This conclusion is confirmed by the data of Hart et al. [31] in studies with Hereford cows. The body weight increases after the 1st week p.p. A calculation supports this observation. A mobilization of 1 kg BW equals 20–21 MJ_NEL_ [71] and a total NEB of 100–200 MJ_NEL_ equals a loss of 5–10 kg BW, which is of minor importance and cannot exactly be determined for a cow with 600–650 kg BW at increasing DMI and a gut fill that probably compensates the tiny loss of BW. The biological NEB during the nutrition of a calf is low, short and “normal” or “usual” and should be related to the current NEB of 2000 MJ or even more for 2 or 3 months. A quantity of 2000 MJ represents a loss of 80 kg of BW, and 1 kg BW means 20–21 MJ_NEL_ [71]. However, replenishment of 1 kg BW requires 25.5 MJ_NEL_ [71]. Milk production via mobilization is not very efficient.

*Milk yield, mobilization and lipidosis:* Milk yield at insufficient DMI induces an NEB, and the metabolic consequences of an NEB are an increase in NEFAs, lipidosis of the liver and BHB production. Glucose is lowered because of the heavy demand by the mammary gland. The three parameters MY, NEB and mobilization are probably genetically correlated and, hence, more MY initiates a deeper and longer NEB with a higher increase in NEFAs and an ectopic deposition of TGs in muscle and the liver (lipidosis). Neither muscle nor the liver is a physiological depot for fat reserves, and the increasing size of fat droplets in the liver cells impairs its functions and triggers the production of BHB during a shortage of glucose. Hence, more milk at insufficient DMI predisposes the cow to lipidosis with subclinical and, finally, with clinical ketosis (Figure 1).

The suggested genetic disposition of ketosis is supported by the moderate heritability (0.3–0.4) for BHB and NEFAs in blood during early lactation [72] and for the fat-to-protein ratio (0.3) in milk during early lactation [73]. NEB and the fat-to-protein ratio are correlated in early lactation [27]. Consistent with the correlation between MY and ketosis is the observation of a genetic correlation (0.49) between milk BHB (≥0.2 mmol/L) and the fat-to-protein ratio of milk in Canadian Holsteins [74].

*Milk yield, ketosis and diseases:* MY has been increased by selection and the prevalence of hyperketonemia (>1.2 mmol/L) is high and has been shown to vary between 11.2% and 36.6% in a study in 10 European countries [75]; this agrees with a compilation of data from the literature in a recent review by McArt et al. [76]. Suthar et al. [75] and McArt et al. [76] have reported an association with a variety of production diseases, in agreement with the data of Ashad and Santos [49] (Table 3) and supporting the conclusion of Berge and Vertenten [77]: “Ketosis (>100 μmol/L in milk, the author) was associated with significant higher odds of all common fresh cow diseases: metritis, mastitis, displaced abomasum, lameness and gastrointestinal disorders”. Furthermore, cows with ketosis (>1.2 mmol/L) exhibit increased inflammatory biomarkers, such as serum amyloid and haptoglobin [78]. These findings indicate a general health risk at BHB > 1.2 mmol/L in blood or its equivalent in milk. Evidently, BHB and lipidosis are associated because the correlation between BHB and the diseases mentioned above is almost identical with the association between lipidosis and such diseases [79,80,81]. An excellent compilation of these problems is presented in the comprehensive review on BHB of Benedet et al. [82].

*Homeorhesis and ketosis:* Milk yield p.p. is regulated by homeorhesis [19] with the priority of “milk secretion allowing them to proceed at the expense of other metabolic processes even to a point that a disease is created” [19] or “even a pathological state” [20]. The homeorhetic priority of milk secretion includes restrictions of other functions that are unimportant at low MY and minor NEB but exacerbated at high MY and severe and long-lasting NEB and at increased NEFAs and BHB at low glucose. Thus, ketosis can be characterized as a homeorhetic disease, as emphasized many years ago by Baird [83]: “The cow will attempt to maintain milk production despite food deprivation and as a result will become ketotic”. “Cows are only susceptible to the disorder (primary ketosis) during early lactation, when the homeorhetic stimulus to lactate is at a maximum” [84]. This conclusion was confirmed by Gross and Bruckmaier [48], who induced NEB by food restriction later in lactation during a phase of positive energy balance; minor changes in metabolites and hormones occurred, together with a reduction in MY. The homeorhetic priority for MY is completed by the end of NEB and the decline in BCS. Possible restrictions of other functions are no longer necessary. The priority is the cow and now, very often, a new pregnancy. In a recent review, hypocalcemia has also been characterized as a homeorhetic disease [85].

*NEFAs, ketone bodies and acceleration of pathophysiology:* A growing body of evidence suggests that metabolites released during NEB or produced by the liver exacerbate the pathophysiological cascade. NEFAs cause a reduction in DMI [86,87], which is also the case for BHB [86], as shown by the negative correlation between BHB and DMI [88]. Thus, an insufficient DMI is further reduced by NEFAs and BHB and will worsen NEB. Additionally, ketogenesis lowers glucose synthesis [40], hyperketonemia in sheep decreases endogenous glucose production [89] and intravenous BHB infusion for 48 h in cows reduces the blood glucose concentration [90]. These interactions make the known reciprocal relationship between BHB and glucose in ketotic cows understandable [91]. Moreover, NEFAs and BHB reduce the expression of GHR-1A and IGF-1 mRNA in liver cells in vitro [91], possibly enhancing the uncoupling of the GH-IGF-1 axis in vivo and possibly explaining the negative correlation between NEFA/BHB and IGF-1 concentrations [92,93].

Ketotic cows exhibit high GH and low IGF-1 concentrations [91], and low IGF-1 before parturition is associated with ketosis p.p. [94]. An uncoupling of the GH-IGF-1 axis appears to be highly likely in ketotic cows, because the treatment of ketotic cows (≥1.3 mmol/L BHB) with 325 mg bSt (bovine somatotropin) on the day of enrolment and at 14 d later is unsuccessful and, thus, “is not recommended as a therapy for cows with hyperketonemia” [95]. The absence of GH effects supports the assumption of an uncoupled GH-IGF-1 axis in ketotic cows.

Taken together, the hormonal and metabolic changes in cows with subclinical ketosis facilitate the creation of a vicious circle, because the insufficient DMI is further reduced by NEFAs and BHB, which finally promote the uncoupling of the GH-IGF-1 axis. The negative consequences of an intact GH-IGF-1 axis for the health of the dairy cow were recognized over 20 years ago by Lucy et al. [96]: “The consequences of inadequate GHR 1A expression are serious. The liver remains unresponsive to GH and various GH-dependent processes (including gluconeogenic mechanism) are not initiated. This may predispose the cow to fat liver and ketosis and preclude the normal hepatic mechanism for nutrient partitioning during increased lactation”. Hence, “optimising the somatotropic axis of IGF-1, growth hormone and growth hormone receptors around calving is a major factor in maintaining health and productivity in early lactation in high yielding dairy cows” [97]. Indeed, the knock-out of the gene for GHR in the liver of mice (GHR KO mice = uncoupling of the GH-IGF-1 axis) initiates hormonal and metabolic changes [98] that resemble the lipidosis and ketosis seen in cows: increased GH and NEFA concentrations in blood, insulin resistance and low IGF-1 concentrations. The lipidosis of the liver in mice is associated with inflammation and oxidative stress independent of a metabolic load, as in dairy cows.

The general pivotal role of IGF-1 on further health functions in cows was demonstrated several years ago. Taylor et al. [99] observed the impairment of fertility in cows at low IGF-1 early p.p., and IGF-1 has long been known to be involved in immunity and inflammation [100,101]. Furthermore, IGF-1 stimulates renal 25-hydroxyvitamin D-1-hydroxylase activity in mice in vivo [102].

In a recent comprehensive review, Horst et al. [37] reject the explanations outlined above and conclude: “It is time to re-evaluate the traditional paradigm of the periparturient dairy cow. The body of evidence linking changes in increased circulating NEFAs, hyperketonemia, and hypocalcemia with negative outcomes has never been overly strong. Further, the doctrine lacks biological plausibility as these are natural homeorhetic adaptations that healthy cows use to synthesize milk, an integral component of the mammalian reproductive cycle. A more likely reason for the observed correlation is that they are merely signs of immune activation.”

The major difference between this statement of Horst et al. [37] and the data presented above is the different view taken of homeorhesis. Horst et al. [37] considered homeorhesis to be a positive reaction for the adaptation of the metabolism for milk production. However, homeorhesis with the priority of MY includes possible restrictions at limited resources. The possible drawback of such a priority has clearly been expressed by Bauman and Currie [19]: “Nature has accorded a high priority to the functions of pregnancy and milk secretion, allowing them to proceed at the expense of other metabolic processes even to the point that a disease state is created” or “even a pathological state” [20]. Priority for milk and the possible restrictions of health functions are thus two sides of one coin.

*Resource allocation and restrictions:* Any increase in production is connected with an increased demand for required resources: nutrition, environment (barn, temperature and humidity) and proper handling by the farm management. As long as the increase in these resources can be fulfilled, raised production with undisturbed health is possible. However, “unless the environment (resources, the author) is being improved, antagonisms between traits will start to develop as soon as production traits are selected” [103]. Hence, appropriate management is a crucial precondition for production and health [104], which cannot be guaranteed p.p. for the dairy cow for one simple genetic reason. The cows do not eat enough and the shortage of the main resource, nutrients, will cause restrictions for other physiological functions and will finally trigger health risks. Hence, the homeorhetic priority for MY at restricted resources will risk the health of the cow.

*Genetic correlation between milk yield and diseases:* In a comprehensive review concerning production diseases in cattle, Emanuelson [105] concluded: “The situation is further complicated by results showing that a genetic antagonism probably exists between production traits and disease resistance”. The interaction between MY, DMI and NEB fits neatly into this early statement and is probably one reason for the genetic correlation between MY and diseases: MY and ketosis [106,107], MY and mastitis [106,107], MY and fertility [108], MY and diminished longevity [109], MY and embryo loss [110], and MY and lameness [111]. Consistent with the correlation between MY and ketosis is the observation of a genetic correlation (0.49) between milk BHB (≥0.2 mmol/L) and the fat-to-protein ratio of milk in Canadian Holsteins [111]. The genetic correlations do not offer a causal explanation, although most of the diseases [112] and early culling [113] occur during 4–8 weeks p.p., suggesting a direct or indirect association with NEB and lipidosis.

These unfavorable genetic correlations between production and health are probably the reason for the concerns expressed in 1991 by Simianer et al. [106]: “If we continue placing almost all emphasis on milk yield and closely related traits and do not include disease traits in the breeding goal, consequences may be detrimental for future health and total economic merit of dairy cows”. This criticism was repeated by Berry 2011 [114]: “Although there were a few exceptions, selection for increased milk production alone without cognizance of other traits is expected to increase the incidence of mastitis, lameness, cystic ovaries, ketosis and metritis” and recently, unequivocally, in 2021, by Brito et al. [115]: “Unfortunately, this progress (in MY, the author) has been accompanied by strong drawbacks, including loss of genetic diversity and deterioration of key biological mechanisms (e.g., health, resilience, robustness, welfare, longevity) in the most common dairy cattle breeds.”

This final statement of Brito et al. [115] is disappointing because the breeding indices have been changed over decades in many countries. For example, the breeding index for Holstein Friesen dairy cows in Germany reduced MY from 100% in 1996 to 36% in 2022 (present) and includes other traits, such as fertility or health, at 18% (mastitis, fertility, metabolism and lameness). Indeed, the trait production has been reduced around the world [116,117].

Undoubtedly, high production is possible under reasonable health conditions [118], although a recent field study in Germany with ca. 86,000 dairy cows and 765 farms has demonstrated the high prevalence of diseases and short life-long production (<3 lactation) [119], both of which have been confirmed in a recent review by Dallago et al. [120], who stated “that the dairy cow longevity has decreased in most high milking-producing countries over time”. This agrees with a statement of Heringstad et al. [121] in their comprehensive review on lameness that “no studies have reported a reduction in the prevalence of lameness over the last 20 y” and that “involuntary culling (in Germany, the author) due to foot and leg problems has increased over time” [122].

The extensive changes of the breeding indices, but with continual increases in MY, appear to the author not to have led to the expected improvement in health. Genetic antagonism between production and functional traits has evident shortcomings and is still a challenge for the health of dairy cows.

*Miscellaneous:* The shortage of DMI can theoretically be compensated by a higher energy density of the diet: 7.00 MJ_NEL_/kg DM is not unusual. This reasonable assumption might not improve the energy balance of the cow, because “high genetic merit animals put the extra energy from the concentrate-based diets into milk, rather than reducing the energy gap” [123,124]. Partitioning of nutrients for milk is more obviously expressed in high-genetic-merit cows. Furthermore, a higher energy density is obtained by increasing starch and decreasing NDF content, which are associated with two drawbacks. First, starch or (even better) its fermentation product in the rumen, propionic acid, reduces DMI intake, which is probably depressed by a signal from hepatic oxidation [125]. Second, the decease in NDF and the increase in starch facilitates the risk of subacute rumen acidosis (SARA) [126], challenging the health of dairy cows [127]. High starch in the diet (27.82% at 29.92% NDF) induces an elevated systemic oxidative status in the blood and increased autophagy in the liver, with the suggestion of enhanced oxidative damage within this organ [128], although these effects of starch are not constantly observed [66]. Hence, the metabolic challenges and health risks of the high-producing dairy cow are probably a mismatch of hormonal and metabolic changes (NEB, NEFAs, BHB and lipidosis) and possible side effects of a digestive disorder.

## 8. Conclusions

The increased MY during the last century is the result of genetic selection for higher production and of proper management with improved feeding and environment. Unfortunately, this progress has not been accompanied by adequate DMI p.p. to parallel the raised milk requirement. A growing body of evidence indicates that the decline in DMI around parturition has a genetic background with a biological function: securing the nutrition of the calf independently of DMI. Thus, dairy cows p.p. experience an NEB that is short and low at an MY based on the sole nutrition of one calf and results in no serious challenge for the health of the mother. Importantly, here, the amount of milk is limited by the appetite of the calf and, hence, the small NEB is “natural”. This limitation has been neglected during the selection for more milk and for frequent milking [129] with the goal of maximal milk production.

The genetic correlation between MY and insufficient DMI causes the now long-lasting and deep NEB. The homeorhetic priority of MY and the resource allocation theory predicts restrictions of other (health) functions during NEB. Hence, homeorhesis for the “physiological state” [19] of MY is a two-edged sword. At low MY, it secures the nutrition of the calf uncoupled from DMI, as is the case in beef cattle and rarely with health problems. Unfortunately, the advantages of uncoupled DMI from MY at low production are turned, with rising MY, into a pronounced NEB with hormonal and metabolic alterations that challenge the health of the cow, because the prioritization of MY causes a subordination of health functions. The missing energy is covered by lipolysis and the release of NEFAs, but, critically, above the metabolic capacity of the cow, leading to the ectopic accumulation of TGs in the liver. Lipidosis is clearly correlated with subclinical and clinical ketosis and regularly precedes ketosis. Hence, the association of ketosis with “production diseases” probably represents an association between lipidosis and diseases. BHB is the “gold standard” of ketosis and mirrors this disease with its impaired function in gluconeogenesis [130], the detoxification of ammonia [131] and urea production [132], but is probably not the cause of the clinical signs of ketosis [133]. The cascade of events (“production diseases”) is practically unknown in beef cattle [134], with each cow only producing milk for the nutrition of one calf. This underlines the disadvantage of selecting dairy cows for higher MY without adequate DMI. This drawback has been realized and many efforts have been made to breed for higher DMI in early lactation [26] improving feed efficiency [135] and targeting BCS at parturition and limiting BCS loss p.p. [136].

The Gordian knot for the dairy cow is the discrepancy between MY and DMI at high MY with the health problems of subclinical and clinical ketosis and, most probably but indirectly, with the “production diseases”, including inflammation, oxidative and endoplasmatic stress, and immunosuppression. In ancient mythology, the problem of the Gordian knot was solved by its being cut with the sword of Alexander the Great. Such a “cut” for the health problems of the dairy cow remains out of sight, although a better understanding of the biological and genetic constitution and the inclusion of this knowledge in the breeding index could pave the way for sustainable milk production. More milk at inadequate DMI with a high incidence of diseases, early culling, increasing mortality and thus short but intense lifelong production will raise public concerns about production in agriculture and is a serious matter of animal welfare. A lower incidence of diseases and traits of energy metabolism (NEB, BCS, DMI and fat-to-protein ratio) should have high priority in the breeding indices during early lactation. Such a change is long overdue, because the deficits in the animal welfare of high-producing dairy cows have clearly been raised in the literature several times in the past [137,138,139].

## Figures and Tables

**Figure 1 animals-13-03097-f001:**
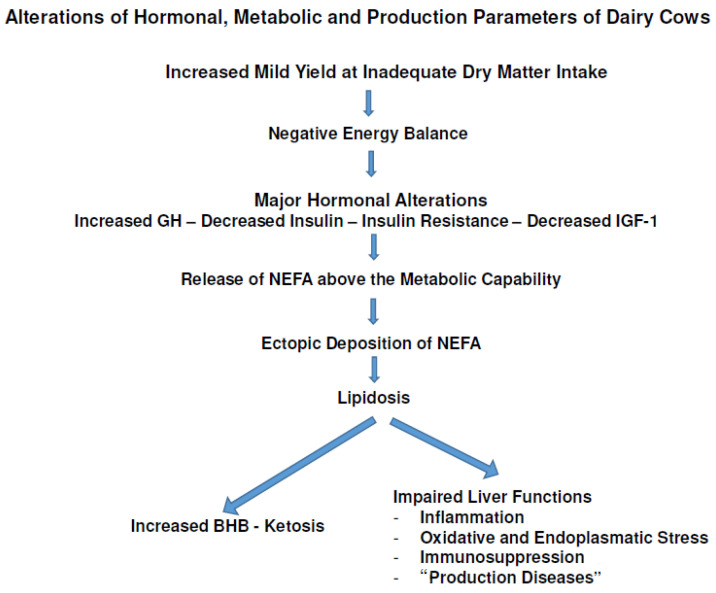
Scheme of alterations of hormonal, metabolic and production parameters of dairy cows. The inadequate DMI causes an NEB with a resulting lipolysis and increased NEFA concentrations for covering the energy deficit. The released NEFAs overwhelm the metabolic capacity of the cow with an unphysiological ectopic deposition in the liver. The subsequent lipidosis causes subclinical or clinical ketosis and generally an impairment of liver functions with the risks of inflammation, oxidative and endoplasmatic stress and immunosuppression. Lipidosis is associated with the so-called “production diseases”. The cascade is accelerated by an uncoupled GH-IGF-1 axis due to the enhanced lipolysis.

**Table 1 animals-13-03097-t001:** Major impacts of growth hormone (GH), insulin, insulin resistance and IGF-1 on the regulation of metabolism postpartum. Adapted from [41,42,43].

Hormone Postpartum	Major Impact on Metabolism
High GH	Lipolysis in subcutaneous and abdominal fat tissue Stimulation of gluconeogenesis in the liver Synthesis of IGF-1 in the liver Growth of mammary gland and synthesis of milk Insulin resistance in muscle and fat tissue
Low insulin p.p.	General: reduction in anabolic metabolism Reduction in lipogenesis in fat tissue Reduced expression of growth hormone receptor 1A in the liver
GH-dependent insulin resistance p.p.	Insulin resistance in muscle and fat tissue Reduced uptake of glucose in muscle and fat tissue Partitioning of glucose to mammary gland No insulin resistance in the liver
Low IGF-1	Lessening of a plethora of para-, auto- and hormonal effects of IGF-1 Reduction in negative feedback mechanism on GH release from the pituitary gland: increase in GH and uncoupling of GH-IGF-1 axis

**Table 2 animals-13-03097-t002:** Alterations of hormonal, metabolic and production parameters of dairy cows with the 33% highest and 33% lowest NEFA concentrations p.p. Adapted from [48].

Parameter p.p.	High NEFA versus Low NEFA
Growth hormone	Higher
Insulin	Lower (not significant)
Insulin resistance	Not significant
IGF-1	Lower
NEFAs	Higher
BHB	Higher
Triglycerides	Higher
Glucose	Lower
Milk	Higher
DMI	Lower
NEB	Lower

**Table 3 animals-13-03097-t003:** Degree of lipidosis associated with the incidence of subclinical and clinical diseases. TG: triglyceride percentage of wet weight: 2.5%, 5% and 7.5% measured at d 8.3 (6–11 d). Diseases were recorded during the first 105 d p.p. Adapted from [49].

Disease (%)	Lipidosis 2.5% TG	Lipidosis 5.0% TG	Lipidosis 7.5% TG
Subclinical ketosis (<1.2 mmol/L BHB)	15.3	24.7	37.5
Subclinical hypocalcemia (<2.0 mmol/L Ca)	30.3	40.8	52.4
Retained placenta	9.3	11.9	15.1
Metritis	12.5	18.2	25.7
Puerperal metritis	6.9	10.2	14.7
Mastitis	14.2	16.9	19.9
Displaced abomasum	1.0	1.0	1.0
Morbidity	36.2	41.3	46.7
Multiple diseases	8.7	13.7	21.1
Survival by day 300	91.1	-	86.3

“Morbidity” includes retained placenta, metritis, displaced abomasum, milk fever, pneumonia and metritis. “Multiple diseases” includes cows with more than one disease listed under morbidity.

**Table 4 animals-13-03097-t004:** Production data and increasing degree of lipidosis. MY and DMI were measured during the first 105 d p.p. TG: triglyceride percentage of wet weight: 2.5%, 5% and 7.5% measured at d 8.3 (6–11 d). Adapted from [49].

Production Data	Lipidosis 2.5% TG	Lipidosis 5.0% TG	Lipidosis 7.5% TG
Milk yield (kg/d)	41.1	42.7	43.2
Dry matter intake (kg/d)	22.0	21.5	21.1
Net energy balance (MJ/d)	−8.65	−16.93	−21.40

## Data Availability

The data used in this review are available in the references.

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
