# Peer review of "Invited Review: Increasing Milk Yield and Negative Energy Balance: A Gordian Knot for Dairy Cows?"

_animals, 2023, doi:10.3390/ani13193097_

Round 1

Reviewer 1 Report

The author provided an excellent review on the relationship between negative energy balance and increasing milk yield. Although numerous review papers related to this topic exist, the author did a great job in addressing this problem by setting the focus on the corresponding dry matter intake. Especially the view on the historical and evolutionary development and genetic selection of dairy cows opens "new" aspects that refer to the issue of a prevailing NEB.

Only some minor issues appeared:

Lines 10/11: Use "gap" instead of "gab" (typo?)

Lines 80 and 87: ... during the transition period

Reviewer 2 Report

Títle: Invited Review: Increasing Milk Yield and Negative Energy Balance: a Gordian Knot for Dairy Cows?

Summary: No comments

Abstract: No comments

Keywords: Write them in alphabetical order

Introduction

Please check the following reference Advances in breeding of dairy cattle (2020) by Julius van der Werf and Jennie Pryce, in which they mention how the dairy cattle selection has evolved from 1917 to 2017 and how other characteristics related to longevity, calving, workability, health, and fertitility have been included in the genetical selection programs since 1997.

It would also be interesting to include the relationship and definition of Dry Matter Intake with Residual Feed Intake (RFI) as a measure of feed efficiency and Feed Saved Australian Breeding Value, variables that are nowadays of interest in livestock production.

I also suggest to check the following literature, which might contribute with important information to the review:

Krattenmacher, N., Thaller, G., & Tetens, J. (2019). Analysis of the genetic architecture of energy balance and its major determinants dry matter intake and energy-corrected milk yield in primiparous Holstein cows. Journal of dairy science, 102(4), 3241-3253.

Xu, W., Van Knegsel, A., Saccenti, E., Van Hoeij, R., Kemp, B., & Vervoort, J. (2020). Metabolomics of milk reflects a negative energy balance in cows. Journal of proteome research, 19(8), 2942-2949.

Li, B., Fikse, W. F., Løvendahl, P., Lassen, J., Lidauer, M. H., Mäntysaari, P., & Berglund, B. (2018). Genetic heterogeneity of feed intake, energy-corrected milk, and body weight across lactation in primiparous Holstein, Nordic Red, and Jersey cows. Journal of dairy science, 101(11), 10011-10021.

Houlahan, K., Schenkel, F. S., Hailemariam, D., Lassen, J., Kargo, M., Cole, J. B., & Baes, C. F. (2021). Effects of incorporating dry matter intake and residual feed intake into a selection index for dairy cattle using deterministic modeling. Animals11(4), 1157.

Manzanilla-Pech, C. I. V., Veerkamp, R. F., Tempelman, R. J., Van Pelt, M. L., Weigel, K. A., VandeHaar, M., & De Haas, Y. (2016). Genetic parameters between feed-intake-related traits and conformation in 2 separate dairy populations—the Netherlands and United States. Journal of dairy science, 99(1), 443-457.

Bilal, G., Cue, R. I., & Hayes, J. F. (2016). Genetic and phenotypic associations of type traits and body condition score with dry matter intake, milk yield, and number of breedings in first lactation Canadian Holstein cows. Canadian Journal of Animal Science, 96(3), 434-447.

I suggest to include an index of the subtopics presented in the Introduction section.

If possible, add a figure describing the “alterations of hormonal, metabolic and production parameters of dairy cows” presented in the review.

Also mention that research including feed intake data is taking place all around the world. However, up to date, there are no results about selection programs with this feature.

Reviewer 3 Report

The authors' hypothesis is that the discrepancy between postpartum milk yield (MY) and dry matter intake (DMI) with subsequent negative energy balance (NEB) has a genetic basis, as a result of selection based solely on MY. Correlations between metabolic variables and reduction in DMI or productive diseases support this theory. The lack of cause and effect relationship between these associations is the main point of criticism, which cannot be ignored. On the other hand, the authors present a plausible hypothesis for these changes, based on the uncoupling of the GH - IGF-1 axis. I think this view is a scientific debate worth encouraging.

The article contains minor errors, described below:

Line 105 and last line of the Table 1– replace “und” with “and”;

Line 121 – correct the quote sign;

Insert a row after table 3 (line 269)

Line 331 – insert measurement unit after the value “-6.5/d”

Line 369 – correct references as follows: Suthar et al. [76] and McArt et al. [77]

Line 498 – reference 119 was not cited.

Reviewer 4 Report

The present review represents the state of the art about our knowledge on the negative energy balance in dairy cows and how this is intrinsically linked to the limited post-partum DMI, particularly in high-producing cows.

The manuscript is in general interesting and well-written, with few minor concerns that preclude its acceptance in the present form.

General comments:

I do not quite understand the "Limitations" chapter. It starts by saying that there is a genetic basis underlying the NEB, so identifying the gap (or limitation, if you want) in our knowledge about how to contrast this (now) pathological status of the high-productive dairy cows. Then becomes like a space where the author addresses the lack of space to talk in detail about the metabolical pathway of the NEB and the linked production disease, giving the reference to the reader where to find this information. Maybe a rephrase of the chapter is needed.

The "Discussion" chapter seems a bit redundant with what has already been addressed in the previous chapters. What is it discussing exactly? In my opinion, the chapter needs to be a little summarized to avoid redundancies and an introduction phrase describing the aim of the chapter should be added.

An accurate revision of the English Language is needed. I suggest the use of a professional English language editing service. Some specific comments are listed below.

L10-11: "gab", I think the author meant "gap" as the gap between the energy input and output that identifies the NEB.

L14: "metabolic capability of the cow", the use of this sentence is present in many parts of the paper. It seems a bit suspended in its context. Capability to do what? Of course, to use the NEFA as energy source. Maybe this explanation is needed to improve the clearness of the sentence. Here and along the manuscript.

L17: "β-hydroxybuteric acid", please change in β-hydroxybutyric acid.

L105: change "und" in "and". Here and in other parts of the manuscript.

L270: I think that the abbreviation TG needs to be defined.

L413: I think that the abbreviation bSt needs to be defined.

Round 2

Reviewer 2 Report

The work is interesting although the writing could be improved and avoid repeating arguments in several of the sections.

It is recommended to see the correlations of several of the variables mentioned on the page https://www.animalgenome.org/cgi-bin/CorrDB/view?tp=tabviw&species=cattle&ctp=phenotypic

Author Response

Please see atteched file.
